# A Novel Method of Wireless Micro Energy Transmission Based on MEMS Micro Coil

**DOI:** 10.3390/mi14111997

**Published:** 2023-10-27

**Authors:** Yongdong Wang, Cheng Yi, Fanxiang Meng, Xuecheng Sun

**Affiliations:** School of Mechanical Engineering, Shanghai University, Shanghai 200444, China; wangyongdong@shu.edu.cn (Y.W.); yicheng@shu.edu.cn (C.Y.); mengfanxiang@shu.edu.cn (F.M.)

**Keywords:** MEMS micro coil, wireless charging, pacemaker

## Abstract

Based on current implantable devices, a battery’s rigidity and large size makes it prone to immune rejection and wound incisions. Additionally, it is limited by its finite lifespan, which hinders long-term usage. These limitations greatly restrict the development of implantable medical device systems towards miniaturization and minimally invasive approaches. Consequently, obtaining high-fidelity and stable biological signals from the target tissue area of the organism remains challenging. Therefore, there is a need to develop wireless power transmission technology. In this paper, we propose a wireless micro energy transfer method based on MEMS micro coils for charging implantable devices. Through simulation calculations, we first investigate the influence of coaxial distance, horizontal displacement, and rotation angle between the MEMS micro coil and the transmitting coil on power transmission. Subsequently, we utilize micro nanofabrication technology to create a MEMS micro spiral copper coil with a line width, thickness, and spacing of 50 µm and a total of five turns. Finally, we conduct wireless power transmission tests on the coil. The results show that, when the transmitting coil and the receiving coil are 10 mm apart and the operating frequency is 100 kHz, the power of the wireless power transmission system reaches 45 µW. This power level is sufficient to meet the power supply requirements of implantable pacemakers. Therefore, this technology holds great potential for applications in the field of wireless power transmission for implantable medical devices, including pacemakers and brain neurostimulators.

## 1. Introduction

Nowadays, people rely heavily on various electrical devices, and charging is a ubiquitous topic when discussing electrical equipment. Wireless charging technology has the potential to overcome some of the limitations and drawbacks of traditional wired charging methods and is an area with great development prospects [1]. For example, wireless charging technology could enable the small and medium power supply for implantable medical devices, eliminating the need for battery replacement surgery when a device’s power runs out, thus increasing the reliability and safety of medical devices [2,3,4].

The application of wireless charging technology in military electronic equipment could also greatly change the application scenario and strategic significance of such equipment [5]. For instance, if miniaturized wireless charging technology is applied to small unmanned detection aircrafts [6], it could enable wireless charging of unmanned aerial vehicle clusters, enriching the corresponding tactics and strategies and making detection much easier. Additionally, wireless charging technology could have an impact on industrial fields and electrical equipment in extreme environments, such as aerospace unmanned detection instruments, deep-sea exploration robots [7], and so on. Thus, the application of wireless charging technology could solve the problem of charging and power supply of electronic detection equipment in special and complex chemical environments [8].

There are currently four main wireless power transfer methods: electromagnetic induction wireless power transfer, magnetic resonance wireless power transfer, mid-range radio frequency radiation wireless power transfer, and ultrasound wireless power transfer. Different power transfer methods have different characteristics. In 2017, Giuseppina et al. proposed an optimized resonant induction link for wireless charging of pacemakers [9]. The proposed link consisted of two coupled planar resonators, and the receiving coil size was about 15 mm × 18 mm. The experimental data showed that the efficiency of the link was about 51%, and it could provide about 61 mW of power. This wireless power transfer method is currently the most mature and has high transmission efficiency due to minimal attenuation in human tissue, but it has the disadvantage of short transmission distance. In 2021, Sevilay et al. discussed wireless energy transfer based on magnetic coupling resonance to deliver power from outside the body to pacemakers [10]. The experimental system worked at a distance of 8 mm between the transmitting coil and the receiving coil, and the operating frequency was 300 kHz to verify the working principle of the proposed WPT charging converter. The final result showed that the WPT efficiency of the system was 82.39% at an output voltage of 4.2 V and an output current of 0.45 A. This wireless power transfer method has the advantage of high transmission efficiency, but it also has the disadvantage of short transmission distance and significant efficiency reduction with the distance between the receiving and transmitting coils. Ref. [11] proposed a system-level implementation of a mid-range radio frequency radiation solution for powering cardiac implants, but there were no experimental results. A miniaturized electric stimulator implant (diameter 2 mm) was constructed, which was much smaller than the implants traditionally used. This solution was used to power the implant in the experiment and demonstrated heart rhythm regulation in an anesthetized rabbit. Mid-range radio frequency radiation is a new emerging deep ultra-low power micro implant power supply solution. The micro implant receiver and functional range of this solution can reach several tens of millimeters, but the receiving power is very low. Song et al. reported an omnidirectional ultrasound design for deep implantation devices (power distance of 20 cm), using a 1 × 5 mm^2^ receiver with a thickness of 1 mm, working at the optimal frequency of 2.3 MHz, and the maximum PTE was 0.4% [12]. At the same time, it provided a total of 2.48 mW of power under load conditions of 5 to 100 kΩ. This method has the advantage of very low transmission efficiency but a long transmission distance, which is suitable for some applications of long-distance transmission in the body.

To apply wireless charging to micro and precision fields, this article introduces the concept of micro coils and simulates the entire wireless charging system. The micro coil is a concept that involves miniaturizing the size of transmitting and receiving coils in ordinary electromagnetic induction wireless charging, especially the receiving coil in this simulation experiment [13]. In this test, the size of the receiving coil is miniaturized to a radius length of about 1 mm, enabling the installation of receiving coils of this size in electronic equipment of almost any size and shape, thus solving the problem of applying wireless charging technology to the field of micro precision [14].

The development of wireless charging technology and its application in related fields is significantly hindered by the problems arising from the use of large transmitting and receiving coils, as well as the low transmission efficiency during wireless charging. Therefore, we recognize the importance of wireless charging technology to environmental sustainable development and consumer electronics development and the significance of wireless charging design based on micro coils. We analyzed and calculated the wireless charging of Micro Electro Mechanical System (MEMS) micro coils through relevant knowledge to find the optimal charging scheme. This article is organized as follows: Section 2 introduces the modeling of the wireless charging system, Section 3 describes the simulation and analysis of MEMS micro coils, Section 4 describes the processing and testing of MEMS micro coils, and the final section is the conclusion.

## 2. System Modeling

The system studied in this article mainly consists of a transmitting coil and a receiving coil, among which the receiving coil is one of the innovative points of this simulation experiment, belonging to the micro coil level. The wireless energy transmission with a frequency of 100 KHz is achieved between the transmitting coil and the receiving coil through electromagnetic induction wireless charging. The parameters for modeling the transmitting and receiving coils are shown in Table 1, and Figure 1 shows the system modeling diagram of the two coils.

The basic principle of electromagnetic induction wireless power transmission technology is based on Faraday’s law of electromagnetic induction and Maxwell’s law, which can be understood as the interaction between magnetic and electric fields. When an AC current is input to the transmitting coil, a magnetic field is generated around the coil. As the electric energy on the transmitting coil changes, the magnetic field it generates also changes. If a receiving coil is placed in the changing magnetic field generated by the transmitting coil, the receiving coil will generate corresponding electrical energy from the changing magnetic field. Therefore, wireless power transmission is achieved in this process.

The transmitting coil used for wireless energy transmission is a commercially purchased coil. In the system modeling, we define the outer diameter of the transmitting coil to be approximately 15 cm, the inner diameter to be approximately 14 cm, and the thickness to be approximately 10 mm. The material used is copper, and in the modeling process, the transmitting part is subjected to AC input at a frequency of 100 kHz through a circuit.

The MEMS coil in the system modeling mainly consists of three layers. The first layer is a bottom coil layer with a thickness of 50 µm, the second layer is a middle connecting pillar layer with a thickness of 50 µm, and the top layer is a left and right connecting pillar layer with a thickness of 50 µm. The entire device is made up of three parts: a substrate of 1 mm thick quartz glass, a support layer of 100 µm thick polyimide, and a coil layer of three-layer structured copper wire.

There are many factors that affect transmission power, among which the mutual inductance M between the receiving coil and the transmitting coil is the most significant. As shown in Figure 2, we can take the schematic diagram of the rotation of the receiving coil around the *x*-axis as an example, where d represents the distance between the receiving coil and the transmitting coil on the *z*-axis; the horizontal offset of the receiving coil is represented by α; the rotation angle of the receiving coil around the *x*-axis is ψ; d$l_{1}$ and d$l_{2}$ are the rotation angle of the receiving coil around the *x*-axis is σ; and two arbitrary points are taken from the transmitting integration units of point A and point B.

The mutual inductance *M* between the receiving coil and the transmitting coil in an electromagnetic induction wireless charging system can be expressed as:(1)$$M=\frac{N_{1}N_{2}\mu_{0}}{4\pi }\int_{0}^{2\pi }\int_{0}^{2\pi }\frac{R_{1}R_{2}(\cos\sigma \cos\psi +\sin\sigma \sin\psi \cos\alpha )}{\left|D_{AB}\right|}d\sigma d\psi$$
(2)$$\left|D_{AB}\right|={\left[{\left(R_{2}\sin\psi -R_{1}\sin\sigma \right)}^{2}+{\left(r+R_{2}\cos\psi \cos\alpha -R_{1}\cos\sigma \right)}^{2}+{\left(d-R_{2}\cos\psi \sin\alpha \right)}^{2}\right]}^{\frac{1}{2}}$$

In the above formula, $N_{1}$ represents the number of turns of the transmitting coil; $N_{2}$ represents the number of turns of the receiving coil; $R_{1}$ represents the radius of the transmitting coil; $R_{2}$ represents the radius of the receiving coil; $\mu_{0}$ represents the permeability in vacuum.

In conclusion, it is easy to find out that, in the electromagnetic induction wireless charging system, there are many factors affecting the transmission power, including the *z*-axis distance between the two coils, the horizontal offset distance of the receiving coil, the rotation angle of the receiving coil around the axis, and other relative spatial position parameters between the two coils.

## 3. Simulation of MEMS Micro Coil

In this section, the simulation and calculation are carried out, and the effects of coaxial distance, horizontal offset distance, rotation angle, and so on regarding the power of the MEMS micro coil are studied. The transmitting coil is subjected to AC input with a frequency of 100 kHz through the transmitting circuit. The load type is an alternating electromagnetic field, and an air domain of 500 mm × 500 mm is set. The receiving coil is modeled using a free tetrahedral mesh, with a maximum element parameter of 800 µm and a minimum element parameter of 20 µm. The analysis types include coil geometry analysis and transient analysis of magnetic field and electric field. Finally, the surface integration of the cross-section of the receiving coil is performed to analyze and calculate the current density on the micro coil cross-section. The size of the micro coil cross-section area is then calculated, and the two values are multiplied to obtain the magnitude of the current on the micro coil.

### 3.1. Coaxial Distance

We can see the magnetic flux of two coils at different coaxial distances in Figure 3. By changing the value of d, which is defined as the distance between the two coils in the coaxial direction, we can study the relationship between the distance in the coaxial direction and the power on the micro coil. In Figure 4a, we can see that when the micro coil is coaxial with the transmitting coil, the greater the distance in the coaxial direction is and the smaller the power on the micro coil. Based on Formulas (1) and (2), we can deduce from theoretical analysis that, as the value of d increases, the value of M decreases. This means that the power transmission between the two coils will decrease. Therefore, in practical use, we should try to minimize the value of d while meeting the requirements in order to improve the power received by the MEMS micro coil. Also, from the results, it can be seen that d has a significant effect on the received power.

### 3.2. Horizontal Offset Distance

To analyze the effect of the horizontal distance of the micro coil moving within its plane on its power, we change the distance r of the micro coil as it moves along the *x*-axis direction. Figure 4b illustrates the correlation between power at different *z*-axis distances and offset distances in the *x*-axis direction.

It can be seen in Figure 4b that, when the distance of the micro coil is offset in the horizontal direction, the power generally exhibits a gradual decrease as the offset distance along the *x*-axis increases. But, when the distance between the micro coil and the *z*-axis direction of the transmitting coil is tiny (e.g., 10 mm and 20 mm in the simulation experiment), the power initially increases and then decreases with the increase in the offset distance along the *x*-axis. During this process, a peak point of the power of the micro coil will be generated, and this phenomenon will become more and more pronounced with the decrease in the *z*-axis distance. When the vertical distance between the two coils is large, the magnetic field intensity around the coil gradually decreases as the offset distance from the center increases. Consequently, the power received by the receiving coil also decreases gradually. However, when the vertical distance between the two coils is relatively close, the magnetic field intensity around the coil increases first and then gradually decreases as the horizontal distance from the offset center increases. The main reason for the increase is that, when the transmitting coil is close to the receiving coil in the process of movement, the magnetic field is denser, resulting in higher power received. Subsequently, as the horizontal distance between the coils increases, the magnetic field around the coil becomes sparser, leading to a decrease in received power.

Therefore, in practical use, when the micro coil is far from the transmitting coil along the *z*-axis direction, in order to increase the current and power of the micro coil, the micro coil should be positioned as close as possible to the coaxial position with the transmitting coil, thereby minimizing the horizontal offset distance of the micro coil and improving the efficiency of wireless charging. When the *z*-axis distance between the micro coil and the transmitting coil is very small, in order to maximize the current and power on the micro coil, the micro coil can be appropriately horizontally offset to approach the peak point of power received.

### 3.3. Rotation Angle without Displacement

The relationship between the power on the micro coil and the rotation angle α can be observed in Figure 4c. Combined with the simulation analysis data of the three stages (d = 30, 40, 50 mm), when the micro coil is at the *z*-axis height of 30 mm, 40 mm, and 50 mm, respectively, and does not shift in the x, y direction, the relationship between the rotation angle along the *x*-axis and its power is seen, as shown in Figure 4c.

It can be concluded that when the micro coil remains stationary in the x, y, and z directions, the relationship between the rotation angle of the plane where the micro coil is located and its power is as follows: If the micro coil rotates around the *x*-axis, the power on the micro coil will decrease as the rotation angle increases. Similarly, based on Formulas (1) and (2), we can deduce from theoretical analysis that, as the deflection angle increases, the value of M decreases. This means that the power transmission between the two coils will decrease. Therefore, in practical use, we should try to minimize the deflection angle while meeting the requirements in order to improve the power received by the MEMS micro coil.

### 3.4. Rotation Angle on x-Direction Displacement

In this part, we explore the effects of the micro coil’s rotation on the *x*-axis, while maintaining a *z*-axis coordinate height of 30 mm and displacing along the *x*-axis. We examine the relationship between the micro coil’s power and the rotation angle α.

Based on the data presented in Figure 4d, we can draw the following conclusions: When the micro coil rotates at an angle around the *x*-axis and experiences displacement along the *x*-axis direction, the power on the micro coil generally shows an initial increase followed by a decrease with the increasing displacement distance along the *x*-axis direction. Notably, we observe a prominent peak in the middle of this process. As the coil rotates and horizontally moves, the magnetic field intensity around the coil gradually increases and then gradually decreases as the offset distance from the center increases. The increase is mainly due to the fact that, when the transmitting coil is close to the receiving coil during the movement process, the magnetic field around the coil is denser, resulting in higher received power and a maximum value. Subsequently, as the coil moves away, the magnetic field around the coil becomes sparser, resulting in a decrease in received power.

Therefore, in practical use, in order to maximize the current and power on the micro coil, the micro coil should be appropriately horizontally offset along the *x*-axis direction to approach the peak point, thereby improving the power received by the MEMS micro coil.

### 3.5. Rotation Angle on y-Direction Displacement

In this part, we will know the relationship between the power on the micro coil and r, the distance of displacement along the negative and positive direction of *y*-axis, when the micro coil rotates around the *x*-axis and maintains the *z*-axis coordinate height d as 30 mm, and Figure 5 illustrates the magnetic flux of two coils at different rotation angles and y-direction displacements.

Based on the data presented in Figure 6a, we can draw the following conclusions: When the micro coil rotates around the *x*-axis and experiences displacement in the negative direction of the *y*-axis, the power on the micro coil initially increases and then decreases with the increasing displacement distance along the negative direction of the *y*-axis. Notably, a maximum peak point appears during this process.

From Figure 6b, we can make the following observations: When the micro coil rotates around the *x*-axis and undergoes displacement in the positive direction of the *y*-axis, its power generally increases initially, then decreases, and finally increases again with the increasing displacement distance. The trend is complex, making it difficult to discern a clear pattern. Additionally, it is worth noting that, when the micro coil rotates around the *x*-axis and experiences the same offset distance in the *y*-axis direction, the power of the micro coil is greater when the offset is carried out along the negative direction of the *y*-axis compared with the positive direction. Therefore, it is advisable to avoid positive displacement along the *y*-axis when the micro coil rotates around the *x*-axis; instead, offsetting the micro coil along the negative direction of the *y*-axis by a certain distance will maximize the power on the micro coil.

## 4. Fabrication and Testing of MEMS Micro Coil

### 4.1. Processing and Fabrication

The MEMS micro coil designed in this article is a planar spiral copper coil, with a width, thickness, and distance between the lines all equal to 50 µm, and a total of five turns. The MEMS process has many similarities with the IC process, including not only basic microfabrication processes such as photolithography, etching, bonding, micro plating, polyimide, etc., but also the advanced microfabrication processes such as anisotropic wet etching, wafer bonding, deep reactive ion etching, sacrificial layer etching, critical point drying, and so on, which are unique to MEMS.

As shown in Figure 7, the MEMS coil preparation process is as follows:

Figure 7a Clean and dry the substrate. Clean the substrate with deionized water using ultrasound for 10 min, then wipe the glass substrate clean with acetone cotton and clean it with ultrasound in acetone for 10 min. Wipe the substrate with alcohol cotton and clean it with ultrasound in alcohol for 10 min. Finally, rinse with deionized water and ultrasound for 10 min. Dry with N2 and then dry in a 60 °C oven to remove any contaminants on the surface of the substrate.

Figure 7b Use a magnetron sputtering machine to deposit a Cr/Cu seed layer with thicknesses of 10 nm for Cr and 150 nm for Cu.

Figure 7c Spin coat a 50 µm thick layer of positive photoresist and dry it.

Figure 7d Expose and develop the photoresist according to the pattern of the first lithography mask (transparent areas). Clean the developed photoresist with deionized water, check for overdevelopment or underdevelopment, and dry it.

Figure 7e Electroplate a 50 µm thick layer of copper for the first lower coil layer.

Figure 7f Immerse the device in acetone solution for 20 min. After removing most of the photoresist on the surface, rinse it twice with deionized water and dry with N2. Then, dry in a 90 °C oven for 2 h.

Figure 7g Spin coat another 50 µm thick layer of positive photoresist and dry it.

Figure 7h Expose and develop the photoresist according to the pattern of the second lithography mask (transparent areas). Clean the developed photoresist with deionized water, check for overdevelopment or underdevelopment, and dry it.

Figure 7i Electroplate a 50 µm thick layer of copper for the second intermediate connection pillar layer.

Figure 7j Repeat the immersion process in acetone solution, rinse with deionized water, dry with N2, and dry in a 90 °C oven for 2 h.

Figure 7k Use RIE (Reactive Ion Etching) to remove excess seed layers.

Figure 7l Spin coat a 110 µm thick layer of polyimide to cover the surface of the copper layer. Cure in a 250 °C oven for 4 h.

Figure 7m Planarize the polyimide layer by using chemical mechanical polishing until the intermediate connection pillars are exposed and there are no visible scratches. Rinse twice with deionized water, dry with N2, and dry in a 90 °C oven for 2 h.

Figure 7n Use a magnetron sputtering machine to deposit another layer of seed layer with thicknesses of 10 nm for Cr and 150 nm for Cu.

Figure 7o Spin coat another 50 µm thick layer of positive photoresist and dry it.

Figure 7p Expose and develop the photoresist according to the pattern of the third lithography mask (transparent areas). Clean the developed photoresist with deionized water, check for overdevelopment or underdevelopment, and dry it.

Figure 7q Electroplate a 50 µm thick layer of copper for the third left and right connection pillar layers.

Figure 7r Repeat the immersion process in acetone solution, rinse with deionized water, dry with N2, and dry in a 90 °C oven for 2 h.

Figure 7s Use RIE (Reactive Ion Etching) to remove excess seed layers.

### 4.2. Some Problems in the Fabrication

The planar spiral micro coil designed in this paper is divided into a three-layer structure, including the bottom coil and electrode layer, the center connecting post, and the lead layer. Among them, copper is used as the coil and conductor material and polyimide is the interlayer insulating material, and the following is a sample drawing of the machining process. As we can see in Figure 8, there are some problems that occurred during processing. Figure 8a has stains on the surface, and the possible cause of this problem may be that the cleaning process is not meticulous. Therefore, we need to be more careful in the cleaning process. After cleaning, we should observe the surface under a microscope to check if it is clean. If it is not clean, we need to clean it again until the surface is clean. Figure 8b has severe scratches on the surface, and the cause may be excessive force applied during the grinding and polishing process. Therefore, in the chemical mechanical polishing process, we need to use finer polishing pads and control the force used to prevent such obvious scratches. Due to the situation where some coils are located at the edge of the plating, the coil widths are inconsistent, with some being thick and some being thin, or all the coil widths are thick, as shown in Figure 8c,d. Therefore, we need to avoid placing coils at the edge of the silicon wafer during the design phase. Additionally, during the plating process, we need to apply current at both ends of the silicon wafer to ensure a more uniform coil width. Figure 9 shows the structure of a well-processed MEMS micro coil during the manufacturing process. Figure 9a shows the situation after the bottom coil and connecting pillar have been electroplated. Figure 9b shows the processed MEMS micro coil, including three devices. Figure 9c displays the well-processed intermediate connecting pillar. Figure 9d displays the line width and spacing of the well-processed coil.

### 4.3. Power Impact Factor Testing

We choose the well-processed coils for the next part of the power test. After testing the inductance resistance of the micro coil, we test the received power of the micro coil in different situations. The effects of coaxial distance, horizontal offset distance, and rotation angle on the received power of the micro coil have been investigated in practical experimental tests. It should be noted that in order to make the test results more significant, we have taken a transmitting coil with more power than in the simulation. In order to verify the proposed wireless micro energy transmission system, the experimental setup is built as shown in Figure 10, and the pads in the experiment are used to change the vertical distance.

The graph in Figure 11a illustrates the output power of the MEMS micro coil at varying vertical distances. The actual test results are similar to simulation, indicating a clear trend: as the vertical distance increases, the received power decreases.

Figure 11b depicts the relationship between power and coaxial distance with different horizontal offset distances of the MEMS micro coil. Based on the actual measurements shown in the figure, it can be observed that the received power of the micro coil initially decreases and then increases with increasing horizontal offset distance. Additionally, it is evident that the power changes more significantly at smaller vertical distances, while the changes become less pronounced at larger vertical distances. This phenomenon can be attributed to the emitting coil generating a larger and denser magnetic field at smaller vertical distances and a smaller and sparser magnetic field at larger vertical distances.

Figure 11c shows the variation of power with the rotation angle of the MEMS micro-coil at different vertical distances. From the actual measurements in the graph, it can be seen that the received power of the micro coil decreases continuously with the increase in the rotation angle of the MEMS micro coil. Moreover, when the angle is small, the power change is significant at the beginning, but when the angle is large, the degree of power change decreases with the decrease in power. This is because, when the angle changes are small, the change in magnetic field intensity is more significant, but when the angle changes are large, the change in magnetic field intensity becomes smaller.

As shown in Figure 11d, the power changes with the horizontal offset distance of the MEMS micro coil at different rotation angles. From the measured results in the figure, it can be seen that, with the increase in the horizontal displacement of the micro coil, the received power first decreases and then slightly increases. At the beginning, as the coil moves away, the magnetic flux density around the coil gradually becomes sparse, resulting in a corresponding decrease in the received power. Then, due to the coil being closer to the transmitting coil, the magnetic flux density around the coil becomes slightly denser, but since the coil has already moved away from the center of the coil, the received power only increases slightly. At the same time, the power change is more obvious at smaller angles and less significant at larger angles. In summary, we can see that the rule of change of the actual test and simulation results regarding the effect of coaxial distance, horizontal offset distance, and rotation angle on the received power of the micro coil are generally consistent; unfortunately, we did not provide the study of other factors or a deeper study in this work, but we will continue these works in subsequent research.

Compared with previous research works, as shown in Table 2, our receiving micro coil in this study has some advantages. In 2015, Monti proposed a wireless power transmission system for implantable medical devices [15]. The size of the receiving coil was 14.9 × 14.9 mm, operating at a frequency of 401–406 MHz. It was able to transmit 1 mW of power when the two coils were 10 mm apart. However, when the horizontal offset distance between the two coils was 5 mm, the received power at the receiving coil decreased by 28%. In 2017, Chen proposed 3D wearable Litz double coils with two identical integrated flexible coils, which could be used in an implantable capsule [16]. When the transmission distance was 5 mm at the working frequency of 13.56 MHz, a power of 21.82 mW was received with the load resistance of 100 Ω. Meng presented a miniaturized implantable receiving antenna for wireless powering of a cardiac pacemaker in 2020 [17]. The size of the antenna was 12 × 12 mm, and the system operated within a wide bandwidth. It could receive 1.8 mW of power when the distance between the coils was 6 mm. In this work, we propose a micro-coil with a size of 1.7 mm × 1.7 mm for wireless powering of miniaturized implantable medical devices. Compared with previous works, our coil is smaller in size and the received power per unit volume is greater. The system operates at 100 kHz and can transmit 45 µW of power when the transmitting and receiving coils are 10 mm apart. Even when the two coils have a horizontal offset distance of 5 mm, the received power at the receiving coil only decreases by 7%. Overall, our design offers a smaller coil size with high power output per unit volume and provides stable power supply for many low-power implantable medical devices, even with a certain degree of misalignment. Paralikar introduced a research system for therapeutic implants using optogenetic tissue interfaces [18]. This system only requires 400 µW of power to operate, while achieving the transformation of neural circuits and effective modulation of neural circuits. Mirbozorgi et al. proposed an inductive link for wireless power transfer to millimeter-sized freely floating implants [19]. The power transfer efficiency and power transmitted to the load at a distance of 14 mm, operating at 60 MHz, were 2.4% and 1.3 mW, respectively. S. Hoa simulated power transmission to the left ventricle of the heart and the cerebral cortex region using two different configurations [20]. In the experimental results, when coupling 500 mW of power to the tissue, the measured power transmitted to the heart coil was 195 µW, and for the brain configuration, it was 200 µW. In comparison, a cardiac pacemaker consumes approximately 8 µW [21], which is well within the requirements. From the previous works mentioned above, we can see that the MEMS micro coil designed in this study has great potential for applications in implantable medical devices, including cardiac pacemakers, brain neurostimulators, and so on.

## 5. Conclusions

In this paper, we study a few influence parameters about the simulation, such as the coaxial height of the micro coil relative to the transmitting coil, the displacement in x direction and y direction, and the rotation of the plane in which the micro coil is located. And in terms of the experimental test, we also study the coaxial distance, horizontal offset distance, and rotation angle on the received power of the micro coil. The corresponding simulation and experimental test results are obtained and analyzed, and the relevant conclusions for the wireless charging system based on the micro coil are summarized as follows.
For the optimization of the wireless charging system with the micro coil and the transmitting coil, the coaxial distance between the micro coil and the transmitting coil should be reduced as much as possible, and also the rotation angle of the plane where the micro coil is located should be reduced as far as possible to improve the power on the micro coil.For the wireless charging system optimization of the micro coil in the horizontal distance offset state, the micro coil should be close to the coaxial position with the transmitting coil as much as possible, that is, the offset distance of the micro coil along the horizontal direction should be reduced as much as possible.When the micro coil rotates at a certain angle around the *x*-axis and deviates along the *x*-axis direction, the offset distance of the micro coil along the *x*-axis direction should be properly decreased as much as possible. When the micro coil rotates around the *x*-axis and the distance displacement in the direction of the *y*-axis occurs, the displacement distance should be reduced properly so that the distance can approach the distance corresponding to the maximum peak point and maximize the power of the micro coil.

In summary, this paper proposes a wireless micro energy transfer method based on MEMS micro coils for charging implantable devices. The main challenge currently faced by MEMS micro coils is the inability to achieve high transmission efficiency, which results in significant energy loss. Our fabrication process needs further optimization, and the geometric dimensions of the coils can also be further optimized to improve the overall transmission efficiency. Therefore, in future work, we hope to adopt MEMS coil arrays, such as connecting multiple small coils in series, to achieve more efficient transmission and reduce energy loss. Additionally, due to the miniaturization and adaptability of this device, this transmission method has great potential for application in future implantable medical devices. Therefore, in future work, we aim to apply these MEMS micro coils to a wider range of implantable medical devices, including low-power devices such as implantable pacemakers, to provide energy supply.

## Figures and Tables

**Figure 1 micromachines-14-01997-f001:**
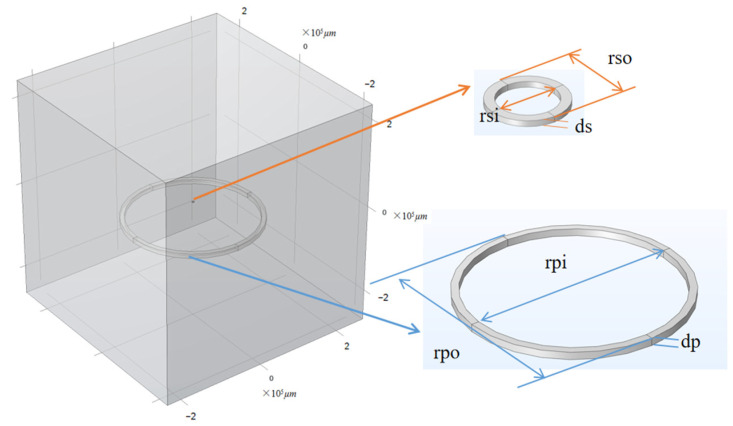
Modeling of wireless energy transfer system with the important structure parameters of the receiving coil and the transmitting coil.

**Figure 2 micromachines-14-01997-f002:**
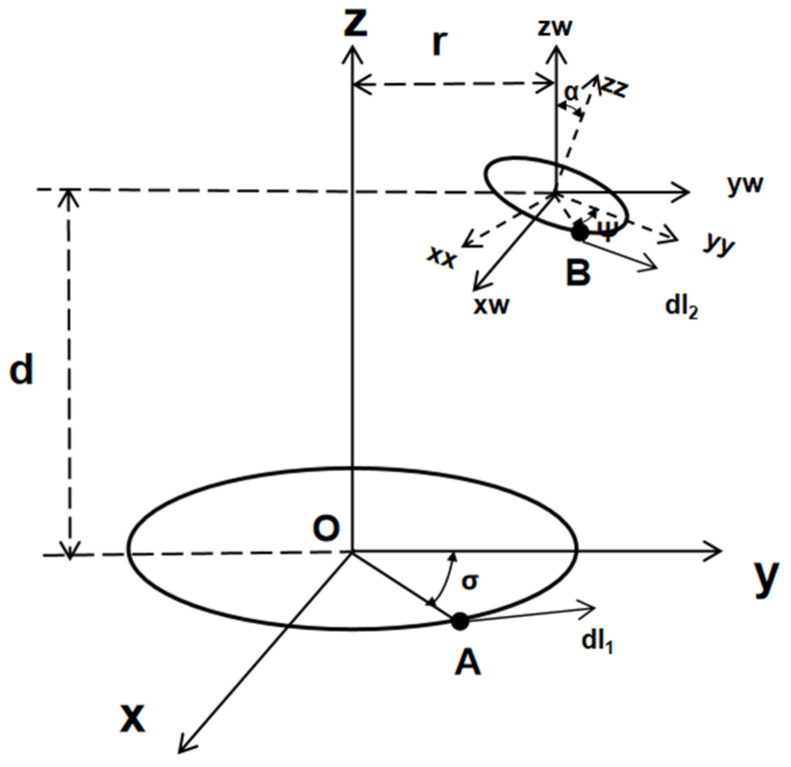
Schematic of the spatial coupling model between the receiving coil and the transmitting coil.

**Figure 3 micromachines-14-01997-f003:**
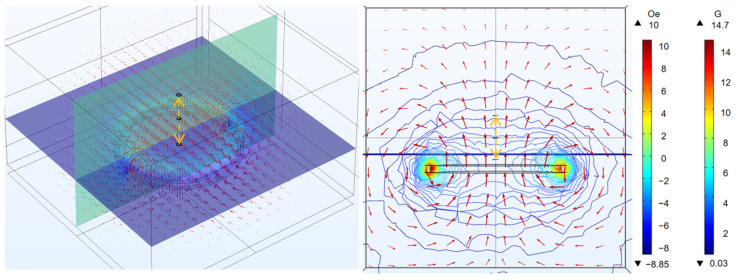
Magnetic flux of MEMS micro receiving coil and transmitting coil at different coaxial distance (Section: magnetic field, z-component (Oe); Section: magnetic flux density (G)).

**Figure 4 micromachines-14-01997-f004:**
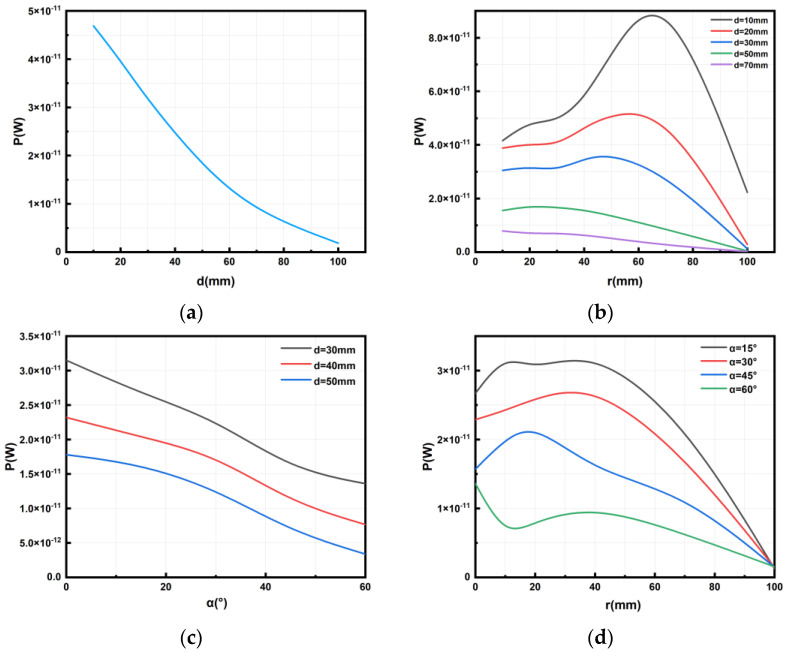
Relationship between power and factors of MEMS micro receiving coil and transmitting coil: (**a**) relationship between power and coaxial distance of two coils, (**b**) relationship between power and the offset distance of the *x*-axis of two coils, (**c**) relationship between power and rotation angle around the *x*-axis without displacement of two coils, and (**d**) relationship between power and rotation angle around the *x*-axis with displacement along the *x*-axis of two coils.

**Figure 5 micromachines-14-01997-f005:**
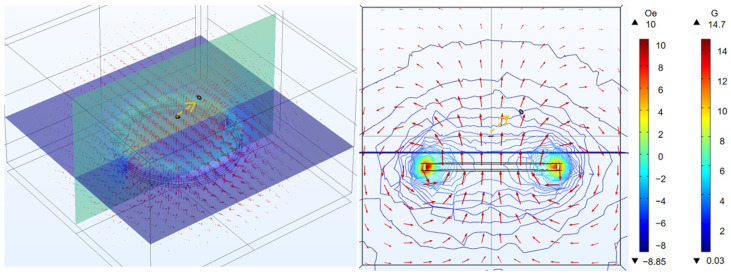
Magnetic flux of MEMS micro receiving coil and transmitting coil at different rotation angle and y-direction displacement (Section: magnetic field, z-component (Oe); Section: magnetic flux density (G)).

**Figure 6 micromachines-14-01997-f006:**
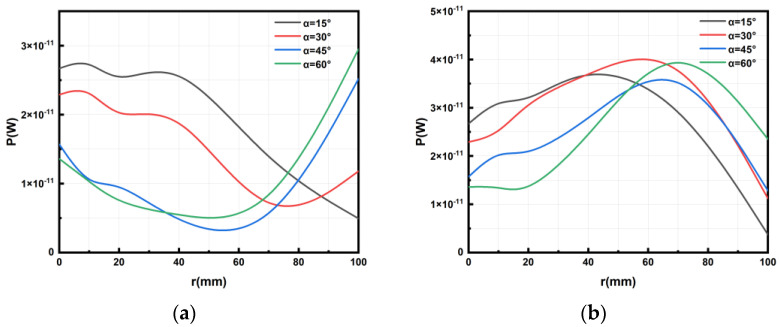
Relationship between power and its offset along the *y*-axis of MEMS micro receiving coil and transmitting coil: (**a**) relationship between power and the positive direction of the *y*-axis; (**b**) relationship between power and the negative direction of the *y*-axis.

**Figure 7 micromachines-14-01997-f007:**
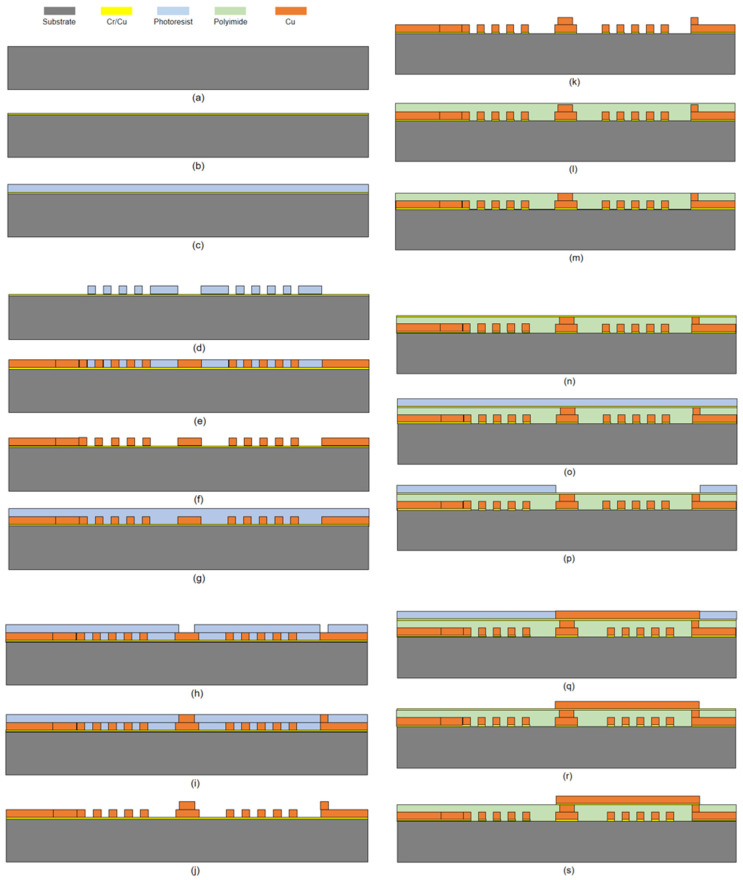
The fabrication process of MEMS micro coil.

**Figure 8 micromachines-14-01997-f008:**
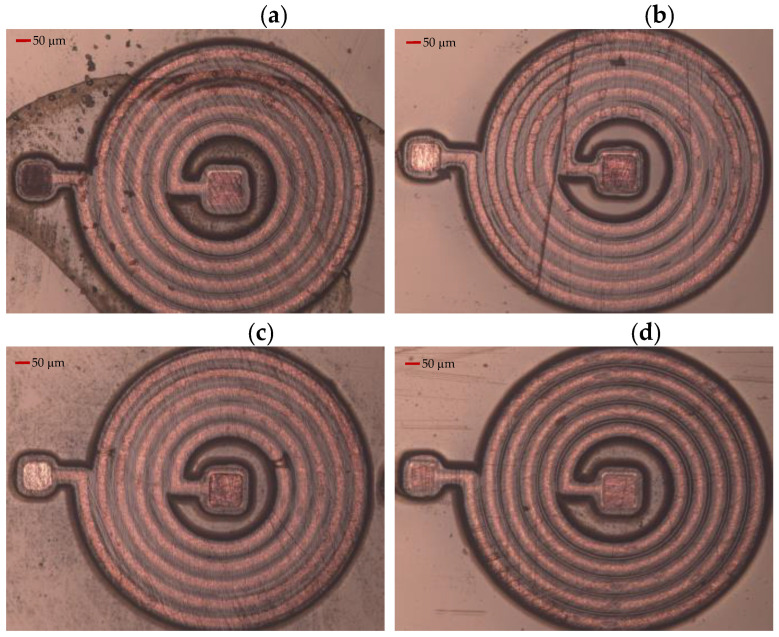
Processed MEMS micro coil with some problems. (**a**) Stained on the surface of MEMS micro coil, (**b**) severe scratches on the surface of MEMS micro coil, (**c**) the widths of MEMS micro coil are inconsistent, and (**d**) the widths of MEMS micro coil are thick.

**Figure 9 micromachines-14-01997-f009:**
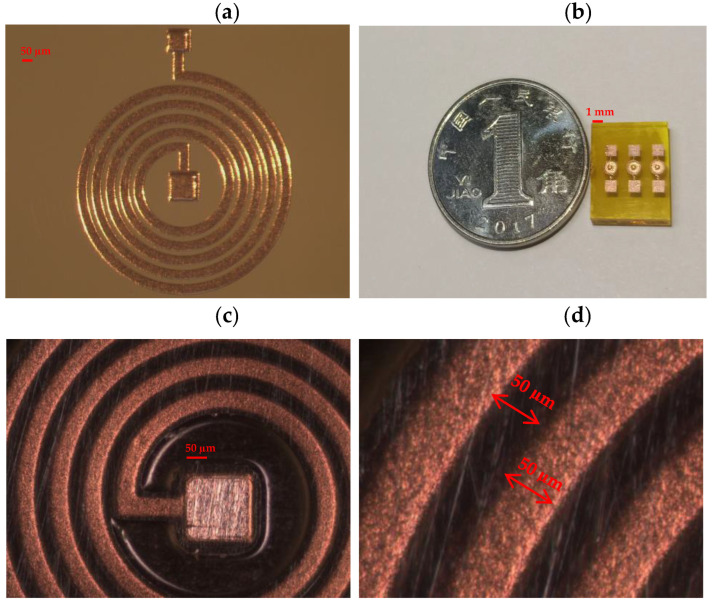
Processed MEMS micro coil with good quality: (**a**) the bottom coil and connecting pillar have been electroplated, (**b**) processed MEMS micro coil with three devices, (**c**) well-processed intermediate connecting pillar, and (**d**) the line width and spacing of well-processed coil.

**Figure 10 micromachines-14-01997-f010:**
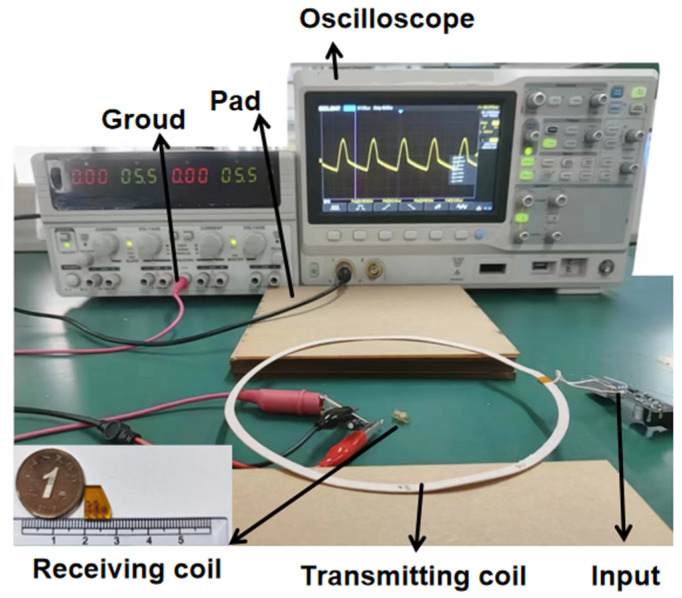
The wireless micro energy transmission system of the experimental setup.

**Figure 11 micromachines-14-01997-f011:**
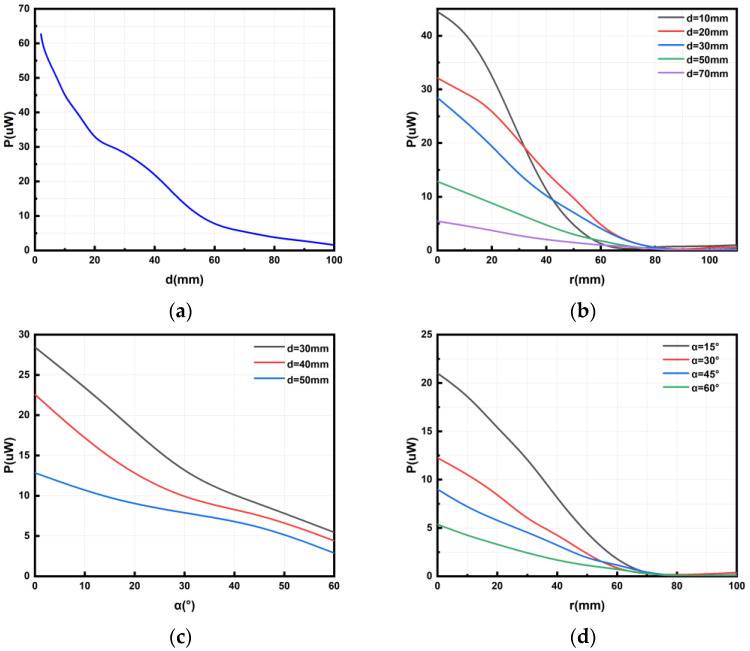
Relationship between power and some factors of MEMS micro receiving coil and transmitting coil: (**a**) relationship between power and coaxial distance, (**b**) relationship between power and horizontal offset distance, (**c**) relationship between power and rotation angle, and (**d**) relationship between power and horizontal offset distance in different rotation angles.

**Table 1 micromachines-14-01997-t001:** The parameters for modeling.

Parameter	Value	Description
rpi	14 cm	Inner diameter of transmitting coil
rpo	15 cm	Outer diameter of transmitting coil
dp	10 mm	Thickness of transmitting coil
Rp	1 Ω	Resistance of transmitting coil
rsi	0.75 mm	Inner diameter of receiving coil
rso	1 mm	Outer diameter of receiving coil
ds	0.15 mm	Thickness of receiving coil
Rs	0.2 Ω	Resistance of receiving coil

**Table 2 micromachines-14-01997-t002:** Comparison with other work.

Year	Authors	F (MHz)	Transfer Distance (mm)	Size (mm × mm × mm)	Power Output per Unit Volume (mW)	Application
2015	Monti et al. [15]	403	10	14.9 × 14.9 × 1.5	0.003	Pacemaker
2016	Chen et al. [16]	13.65	5	7.3 × 7.3 × 7.3	0.056	Implantable capsule
2020	Wang et al. [17]	272–1504	6	12 × 12 × 0.635	0.019	Pacemaker
2023	This paper	0.10	10	1.7 × 1.7 × 0.15	0.103	Pacemaker

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
