# Peer review of "A Novel Method of Wireless Micro Energy Transmission Based on MEMS Micro Coil"

_micromachines, 2023, doi:10.3390/mi14111997_

Round 1
Reviewer 1 Report
Comments and Suggestions for Authors
This manuscript presents a wireless micro energy transmission method based on MEMS micro coil. This work reports the influence of the coaxial distance, horizontal displacement, and rotation angle between the transiting coil and MEMS micro coil on power transfer. This manuscript can be enhanced by considering the following issues:
1.- The abstract section should include an introduction on the research problem. In addition, this section could add the main data of the materials and fabrication process used to fabricate the MEMS micro coil. The units microWatts must be revised in the abstract.
2.-The introduction sections should incorporate more discussions on the main advantages and drawbacks of the different wireless charging technologies reported in the literature. Furthermore, the authors must consider the main scientific contribution or innovation of their proposed device compared to other technologies reported in the literature.
3.- The description of the system modeling (third section) is weak and short. This section must be improved including more information and figures of the working principle, main components and materials of the MEMS micro coils.
4.- The resolution and quality of figures 1 and 2 is weak. The captions of all the figures must be improved by considering more information on the description of all the figures.
5.-All the parameters used in the Equations must be described in the manuscript.
6.- In the third section should include other title. The title "Calculation and analysis simulation" is not suitable. In addition, this section must consider more descriptions on the FEM models, including the boundary conditions, load types, mesh, analysis type, and so on.
7.- The resolution and quality of figures 3-6, and 12 can be enhanced. The captions of figures 3,4,5,6, and 12 must be improved by considering more information of the MEMS micro coils.
8.- The discussions of the main results of figures 3-6, and 12 can be improved.
9.- The description of the fabrication of MEMS micro coil should be enhanced by considering more detailed information.
10.- The labels of Figure 8 are small. The caption of this figure must be enhanced.
11.- The caption of figures 9 and 10 must be improved. The scale line of figure 10 is unclear.
12.- The authors can include discussions on the reliability of the proposed MEMS micro coil.
13.- What are the main challenges of the MEMS micro coil?
14. What are the future research works?
Comments on the Quality of English Language
The English grammar can be improved.
Reviewer 2 Report
Comments and Suggestions for Authors
The research presented by Wang et al. introduces an innovative approach to wireless micro energy transmission leveraging the capabilities of a MEMS micro coil. Central to their exploration is the potential use of this technology to charge implantable medical devices, notably pacemakers and brain neurostimulators. Overall, the findings are of significant interest to the readership of the "Micromachines MDPI" journal and merit consideration for publication, contingent upon certain enhancements.
Recommended Improvements:
1) In the sentence "For this, first-principles methods based on DFT theory (structural and elastic) and quasi-harmonic Debye model (thermodynamics) were used.", "with" should be replaced with "in comparison to" for better clarity.
2) "45uw" should likely be "45 µW" to correctly represent the unit of microwatts.
3) In "The application of wireless charging technology in relevant military electronic equipment will also greatly change the application scenario and strategic significance of the equipment [5].", the phrase "strategic significance of the equipment" might be clearer as "strategic significance of such equipment".
4) In "The problems caused by enormous transmitting coil and receiving coil and the low transmission efficiency happened while charging wirelessly hinder the development of wireless charging technology and the application in related fields greatly.", the structure is a bit convoluted. Consider rephrasing for clarity.
5) Under topic 4, it would be beneficial to provide more specific details regarding the equipment utilized in the microfabrication process. This includes information like the manufacturer, model, and any pertinent experimental details. Consider presenting this data in a supplementary document for comprehensive understanding.
Comments on the Quality of English Language
Grammar and Syntax: There are instances where sentence construction is complex, making the content challenging to comprehend immediately. Simplifying and restructuring some sentences can improve clarity.
Terminology and Units: Some notations, like "45uw", need correction to their standard form, "45 µW", to maintain consistency and accuracy throughout the manuscript.
Round 2
Reviewer 1 Report
Comments and Suggestions for Authors
The authors improved their manuscript based on the reviewer's comments.
Comments on the Quality of English LanguageThe English grammar is good.